# Detection of Cd^2+^ in Aqueous Solution by the Fluorescent Probe of CdSe/CdS QDs Based on OFF–ON Mode

**DOI:** 10.3390/toxics10070367

**Published:** 2022-07-03

**Authors:** Dengpeng Wang, Feng Gao, Xianran Wang, Xiaomei Ning, Kaituo Wang, Xinpeng Wang, Yuezhou Wei, Toyohisa Fujita

**Affiliations:** 1Guangxi Key Laboratory of Processing for Non-Ferrous Metals and Featured Materials, MOE Key Laboratory of New Processing Technology for Non-Ferrous Metals and Materials, Guangxi University, Nanning 530004, China; 1915391031@st.gxu.edu.cn (D.W.); 2015301071@st.gxu.edu.cn (X.W.); 2115391053@st.gxu.edu.cn (X.N.); wangkaituo@gxu.edu.cn (K.W.); wangxinpeng@gxu.edu.cn (X.W.); 2School of Resources, Environment and Materials, Guangxi University, Nanning 530004, China; 3School of Nuclear Science and Technology, University of South China, Hengyang 421001, China; yzwei@gxu.edu.cn

**Keywords:** QDs, fluorescence quenching, fluorescence restore, Cd^2+^ detection, adsorption, desorption

## Abstract

The detection of heavy metals in aqueous solutions has always attracted much attention from all over the world. A fluorescent probe of CdSe/CdS core-shell quantum dots (QDs) was designed to detect trace Cd^2+^ in aqueous solutions using the OFF–ON mode rapidly and efficiently, likely based on adsorption and desorption reactions between ethylenediaminetetraacetic acid disodium salt (EDTA) and CdSe/CdS QDs. In the OFF mode, the optical shielding function of EDTA results in fluorescence quenching owing to the strong adsorption ability of EDTA with Cd^2+^ on the sites of CdSe/CdS QDs surface. In the ON mode, the introduction of Cd^2+^ promotes the desorption of EDTA from the EDTA-CdSe/CdS QDs and restores the fluorescence intensity. There were two linear response ranges which were 0.1–20 µmol/L and 20–90 µmol/L for the EDTA-CdSe/CdS system to detect Cd^2+^. The detection limit was 6 nmol/L, and the standard deviation was below 4% for the detection of Cd^2+^ concentration in tap water.

## 1. Introduction

Water pollution includes heavy metal pollution [1,2,3] and radionuclide pollution [4,5]. Due to the non-biodegradable nature, heavy metal pollution has always been a problem, until now. The influence of Cd^2+^ pollution should not be underestimated [6]. As a heavy metal ion, Cd^2+^ has a long biological half-life period of 20–30 years and is accumulated in the human body via polluted water, air, soil, or other ways, causing many diseases of the kidney, liver, heart, lung, or other organs. Studies have shown that it can cause a series of health problems, including several fatal diseases such as diabetes, cancer, and chondropathy, even if the accumulated Cd^2+^ content in the body is very low [7].

Currently, there are several effective techniques to detect Cd^2+^: atomic absorption spectrometry (AAS) [8], atomic fluorescence spectrophotometry (AFS) [9], inductively coupled plasma mass spectrometry (ICP-MS) [10], the electrochemical method [11,12], and the fluorescence probe method [13,14,15]. Compared to the fluorescent probe method, AAS cannot be used for simultaneous analysis of multiple elements, as AFS and ICP-MS require expensive and complex instruments, complex sample preparation, and the electrochemical method, which has the disadvantage of poor selectivity.

The biggest advantage of the fluorescent probe method is its rapid response, visibility, and high sensitivity. In addition, low cost, simple operation, and a wide linear range for detecting heavy metal ions are also obvious advantages. These advantages make for attracting much attention from researchers, and they have been widely used in the biomedical and analytical chemistry fields [16,17].

Fluorescence probes can be divided into organic fluorescence probes and inorganic fluorescence probes [18]. Inorganic quantum dots (QDs) have been widely used in fluorescence detection in recent years on account of the following merits: high fluorescence quantum yield, size-tunable fluorescence emission spectrum, and visibility. There were several reports about fluorescent QDs probes for detecting Cd^2+^, such as CdX (X = Te, Se, S) QDs, ZnS QDs, C QDs, and Au QDs [19,20,21,22]. According to the spectral characteristics, some QDs fluorescent probes are based on the intensity change of a single fluorescence peak, while others are ratiometric fluorescent probes based on the relative intensities of double emission peaks. According to the structure of the QDs, the probes have single crystal, core-shell, or composite crystal [23,24,25]. Usually, the detection of Cd^2+^ by QDs has two methods, i.e., TURN–OFF and OFF–ON [16,26]. So far, a considerable number of defects were caused by the TURN–OFF mode. In comparison, there are few detections from OFF–ON mode. However, many results show that the OFF–ON mode is more accurate than the ON–OFF mode in detecting Cd^2+^ [23].

In this work, a novel QDs fluorescence probe based on OFF–ON mode was developed. Considering that the single-core QDs have many defects on their surface which can affect the luminescence efficiency, CdSe/CdS QDs were designed and prepared in a core-shell structure. A schematic diagram of Cd^2+^ detection in aqueous solution is shown in Figure 1. Under UV excitation, an obvious emission band of 500–750 nm was observed in the photoluminescence emission spectrum of the CdSe/CdS QDs. EDTA molecules were partially dissociated into anions and cations when they were added in the aqueous solution of CdSe/CdS QDs, and then some of the EDTA^−^ ions were adsorbed on the surface of QDs through electrostatic force between EDTA^−^ and Cd^2+^, shielded the fluorescence excitation and emission energy, thus resulting in the quenching of fluorescence emission. At that moment, the detecting system turned out to be the OFF mode. When Cd^2+^ was added, the EDTA^−^ ions chelated with the Cd^2+^ and reduced the number of EDTA^−^ in the solution system. In order to maintain the chemical balance, some EDTA^−^ ions would desorb from the surface of the CdSe/CdS QDs and were released into the solution again, leading to their fluorescence restoration.

## 2. Materials and Methods

### 2.1. Apparatus and Reagents

Fluorescence spectra were taken on a FL3C-111 TCSPC luminescence spectrometer equipped with a 20-kW xenon discharge lamp as a light source (Horiba, Kyoto, Japan). X-ray powder diffraction (XRD) spectra were taken on a Rigaku D/MAX2500V X-ray diffractometer (Rigaku, Tokyo, Japan). The microstructural features of the samples were characterized by a F200X (Semerfeld, Seattle, WA, USA) transmission electron microscope (TEM).

Thioglycolic acid (TGA), NaBH_4_, CdCl_2_·2.5H_2_O, Na_2_S·9H_2_O, and ethylenediaminetetraacetic acid disodium salt (EDTA) in analytical grade and various standard solution of metal ions (K^+^, Na^+^, Mg^2+^, Ba^2+^, Al^3+^, Mn^2+^, Fe^3+^, Ca^2+^, Hg^2+^, Pb^2+^, Cu^2+^, Ag^+^, Ni^+^, Zn^2+^, Cd^2+^) were purchased from Sinopharm Chemical Reagent Co., Ltd. Analytically pure tris (hydroxymethyl) methyl aminomethane (Tris) was purchased from Aladdin company. Ultra-pure water was used in the whole experiment except for particular notations in text, and the resistivity of ultra-pure water is above 18 MΩ·cm (25 °C).

### 2.2. Synthesis of CdSe/CdS Core-Shell QDs

The fabrication method of CdSe/CdS QDs has already been reported elsewhere [27]. In brief, a certain amount of Se, NaBH_4_, and 10 mL ultrapure water were added into a three-necked flask under the N_2_ atmosphere, and then stirred vigorously until the solution became colorless and clarified. The solution at this time was a NaHSe solution, which would be used as a precursor of Se for the next step. A certain amount of CdCl_2_ was dissolved in 100 mL of ultrapure water, and then a certain volume of TGA solution was dropped into it, and the solution changed from colorless to cloudy rapidly. The molar ratio of Cd:Se:TGA in the reaction system was 1:0.5:2.5. The CdCl_2_ solution became clear again when the pH value of the solution was adjusted to 11 with 1 mol/L NaOH solution, and before it, N_2_ had been introduced for 30 min to exclude oxygen. The prepared NaHSe solution was quickly transferred to the CdCl_2_ solution, and the mixture was stirred vigorously and heated under 80 °C for 30 min under N_2_ atmosphere to get CdSe solution. After cooling to room temperature, a certain amount of CdCl_2_ solution and Na_2_S solution were prepared according to the molar ratio of CdSe:CdS = 1:1. The CdSe solution was added drop by drop under intense stirring, and the reaction system was heated to 80 °C and refluxed for 30 min. The final solution obtained by the above process was an orange-red solution, which was washed with anhydrous ethanol, centrifuged 3 times, and subsequently dispersed in ultrapure water. The solution is an as-prepared TGA-capped CdSe/CdS QDs solution.

### 2.3. Fluorescence Quenching Method of CdSe/CdS by EDTA

For the study on the fluorescence quenching of CdSe/CdS QDs by EDTA, the following series of solutions were prepared, i.e., 300 μL CdSe/CdS QDs solution, 2.4 mL Tris-HCl buffer (10 mmol/L, pH = 8.0), 300 μL EDTA with various concentrations. All these solutions were added into a colorimetric dish in turn to form a 3 mL solution system. After incubation for 10 min, the photoluminescence (PL) spectra of the solution system were tested. Fluorescence quenching degree is expressed by *I*/*I*_0_, in which *I* and *I*_0_ represent the PL intensities of QDs with the various concentrations of EDTA and without EDTA, respectively.

### 2.4. Fluorescence Restore Method of EDTA-CdSe/CdS by Cd^2+^

In order to investigate the effect of Cd^2+^ on the fluorescence intensity of the EDTA-CdSe/CdS system, the following series of solutions were prepared: 300 μL CdSe/CdS QDs solution, 2.1 mL Tris-HCl buffer, 300 μL EDTA, and 300 μL Cd^2+^ aqueous solution with various concentrations were added into a colorimetric dish in turn to form a 3 mL solution. After incubation for 10 min, the PL spectra of the solution system were tested. The fluorescence restoring effect is expressed by *I*/*I*_0_, in which *I* and *I*_0_ represent the PL intensities of EDTA-QDs with the various concentrations of Cd^2+^ and without Cd^2+^, respectively.

All PL spectra were measured under the same conditions: the excitation and emission slit were set to 3 nm, and the excitation wavelength was set to 397 nm. The monitoring emission range was 420–780 nm. The fluorescence intensity values were not corrected for inner-filter effects.

## 3. Results and Discussion

### 3.1. Characterization of CdSe/CdS QDs

Figure 2a shows the X-ray diffraction (XRD) pattern of the CdSe/CdS QDs. Three wide diffraction peaks centered at 2*θ* = 25.8°, 43.2° and 50.5° were observed, corresponding to the characteristic peaks of (111), (220) and (311) lattice planes of cubic CdSe or CdS. There were mutual stresses between the core of CdSe and the cell of CdS that caused the lattice parameters to change, and thereby the shifts of the diffraction peaks happened specifically for the two peaks at a higher angle. A similar diffraction pattern was observed in CdSe/CdS nanoparticles in which CdS was epitaxially grown on a CdSe core [28]. These obvious wide peaks reflect the basic characteristics of nanoparticles. The micro-morphology of TGA-capped CdSe/CdS QDs is shown in Figure 2b,c. These QDs, with a nearly spherical shape, display a good dispersion property. The insert graph in Figure 2c shows the HRTEM (high-resolution transmission electron microscopy) image obtained by focus on a nanoparticle within the view field and the spacing of the neighbouring lattice fringe is 0.35 nm, corresponding to the (111) lattice plane of CdSe, which further proves the final prepared products in this work to be CdSe/CdS nanoparticles. By counting the particle size of all the nanoparticles in Figure 2c, the size distribution of the QDs is shown in Figure 2d. The particle size is mainly in the range of 4–20 nm, and the average particle size is about 12 nm.

### 3.2. Fluorescence Quenching Effect of EDTA on CdSe/CdS QDs

The introduction of EDTA can effectively reduce the PL intensity of CdSe/CdS QDs, as shown in Figure 3a. There is a broad emission band centered at about 600 nm for all the CdSe/CdS QDs systems with various EDTA concentrations. As the concentration of EDTA increases, the PL intensity of the EDTA-CdSe/CdS QD system decreases gradually, with a red shift of the emission band in the spectra. This is because EDTA was chemically absorbed on the surface of QDs, which caused QDs to cluster [29]. The change trend of PL intensity can be expressed by the relationship between *I*/*I*_0_ and the concentration of EDTA in Figure 3b, in which *I* and *I*_0_ represent the PL intensities of QDs with the various concentrations of EDTA and without EDTA, respectively. It can be seen that the PL intensity decreases gradually with the increase of EDTA concentration. The fluorescence quenching is fast when the EDTA concentration is less than 35 µmol/L, but becomes slow when the EDTA concentration is more than 35 µmol/L. This is because some Cd^2+^ sites on the surface of CdSe/CdS QD were occupied with the increasing concentration of EDTA, and the more the combination between EDTA and CdSe/CdS QD becomes difficult, the more the fluorescence quenching slows down. The fluorescence intensity response curve of CdSe/CdS QDs can be divided into two stages, as shown in Figure 3c,d, respectively. In the concentration range of 0–35 µmol/L EDTA, the linear relationship between *I*/*I*_0_ and the EDTA concentration (*C*_EDTA_) can be expressed by Equation (1),
(1)$$I/I_{0}=0.98788+(-0.02182)C_{\mathrm{EDTA}}$$
and the correlation coefficient (R^2^) of Equation (1) is 0.994. In the range of 35–60 µmol/L EDTA, the correlation equation can be expressed as Equation (2),
(2)$$I/I_{0}=0.39829+(-0.00582)C_{\mathrm{EDTA}}$$
and the correlation coefficient (R^2^) is 0.976.

As a common metal chelating agent, EDTA will be chemically adsorbed on the surface of QDs to chelate with Cd^2+^ sites of CdSe/CdS QD when it was added to the CdSe/CdS QD solution, and a large area of optical active sites on the surface of these fluorescent CdSe/CdS QDs were masked, resulting in the fluorescence quenching. It caused a blue shift in the absorption spectra, as shown in Figure 4. After the introduction of Cd^2+^, the absorption peak was red shifted, which was due to the partial leakage of the photon into the shell matrix [30].

In order to further explore the quenching mechanism of QDs by EDTA, temperature experiments and measurement of the fluorescence lifetime were conducted. The results are shown in Figure 5. The fluorescence quenching data were analyzed by Stern–Volmer Equation (3) [29],
(3)$$I_{0}/I=1+K_{\mathrm{sv}}[Q]$$
where *K_sv_* is the quenching constant and [*Q*] is the concentration of quenching agent. As shown in Figure 5a, when the temperature is 298 K and 308 K, the relationship between *I*_0_*/I* and *Q* conforms to the Stern–Volmer equation. The value of slope decreases with the increase of temperature, indicating that static quenching occurs between EDTA and QDs. As shown in Figure 5b, the fluorescence lifetimes of QDs, QDs + 10 µmol/L EDTA and QDs + 20 µmol/L EDTA are 27.3 ns, 24.4 ns and 23.7 ns, respectively. The change of the lifetime is not obvious with the increase of Cd^2+^ concentration, consistent with the characteristics of static quenching. When the temperature is 298 K and 308 K, the quenching constants *K_sv_* were calculated to be 4.49 × 10^4^ L/mol and 3.82 × 10^4^ L/mol. According to the Formula (4) [31],
(4)$$K_{q}\tau_{0}=K_{\mathrm{sv}}$$
where, *τ*_0_ is the lifetime of CdSe/CdS QD, the bimolecular quenching rate constant (K_q_) is 1.64 × 10^12^ L·mol/s and 1.39 × 10^12^ L·mol/s, respectively, much higher than the maximum dynamic quenching rate 2.0 × 10^10^ L·mol/s. Therefore, it can be concluded that the quenching mechanism of the QDs by EDTA belongs to a static quenching process.

The concentration of EDTA has great influence on the sensitivity of EDTA-CdSe/CdS QDs to Cd^2+^ detection when Cd^2+^ concentration lies in a proper range. The quenching effect is not obvious when the concentration of EDTA is too low, while the detection of Cd^2+^ was not accurate when the concentration of EDTA is excessive. The fluorescence quenching efficiency (1 − *I*/*I*_0_) for the CdSe/CdS QDs system reaches 90% when 50 µmol/L EDTA is added. Furthermore, the quenching efficiency of CdSe/CdS QDs increases to 99.5% when the EDTA concentration increases to 60 µmol/L, meaning that the fluorescence is almost completely quenched. For comparison, 50 µmol/L EDTA was selected for subsequent fluorescence restoration experiments to detect Cd^2+^.

### 3.3. Relationship between Fluorescence Intensity and Incubation Time and pH of Solution

The fluorescence stability of CdSe/CdS QDs, the quenching rate of CdSe/CdS QDs by EDTA, and the fluorescence restoration efficiency for the EDTA-CdSe/CdS system by Cd^2+^ were determined by the changes of PL intensity of these three systems over time. The experimental results are shown in Figure 6. The fluorescence intensity of CdSe/CdS QDs remains stable with the prolongation of incubation time, indicating a good fluorescence stability for the QDs system. After the addition of EDTA, the fluorescence quenching of CdSe/CdS QDs is very significant, and the PL intensity begins to be stable within 5 min, indicating that the reaction between EDTA and CdSe/CdS QDs is rapid, and that the fluorescence quenching is very effective. After Cd^2+^ was introduced into the EDTA-CdSe/CdS QDs system, the fluorescence began to be restored, and the fluorescence was almost completely restored within 3 min, remaining stable after that.

The pH in the solution system can affect the fluorescence intensity of QDs, as well as the sensitivity and selectivity of detected substances [32]. Figure 7a shows the influence curve of solution pH on PL intensity of TGA capped CdSe/CdS QDs. When the pH increases from 5.5 to 8.0, the PL intensity of the QDs increases gradually, and after that, the PL intensity tends to be stable with continual increase of pH value. This is because the mercaptan groups of TGA capped QDs are not stable under acidic conditions, which enhances the direct contact frequency between the QD surface and the aqueous solution. At this time, the fluorescence is weaker. Figure 7b shows the effect of solution pH on the fluorescence quenching of TGA capped CdSe/CdS QDs induced by EDTA, and on the fluorescence restoration for the EDTA-CdSe/CdS QDs system by Cd^2+^. With the increase of the pH value, *I*/*I*_0_ first decreases and then increases until the solution pH reaches 9.0, and at last decreases again to a higher pH range. The mechanism of the solution pH on the fluorescence intensity is very complicated. As a weak acid, the dissociation equilibrium constant of EDTA becomes smaller with the decrease of the solution pH value from 7.5 to 6, thus reducing the total number of EDTA^−^ ions in the solution, which causes more ions of EDTA^−^ to desorb from the surface of the fluorescent QDs and re-enter into the solution. This leads to a weaker quenching effect of EDTA, thus enhancing the fluorescence restoration rate of the QDs system in a lower pH solution by the introduction of Cd^2+^. This explains why *I*/*I*_0_ decreases with the increase of pH values in the range of pH 6–7.5. However, just like the discussion regarding the results of Figure 5a, higher pH is benefit for the stability of the mercaptan groups of TGA. The competition between these two factors determines the increasing change trend of fluorescence intensity in pH 7.5–9.0. However, if the solution pH value becomes too high and surpasses 9.0, Cd^2+^ in the solution system tends to react with OH^−^ to form Cd(OH)_2_ precipitation [30], and thus *I*/*I*_0_ begins to once again decrease.

### 3.4. Detection of Cd^2+^ in Ultrapure Water Solution

The absorption of EDTA on the surface of CdSe/CdS QDs produced a shielding function for the optical absorption and emission of QDs, which resulted in fluorescence quenching for the QDs system. Then Cd^2+^ was introduced to restore fluorescence of EDTA-CdSe/CdS QDs, and the restoration efficiency depends on the Cd^2+^ concentration in the detected water sample. As shown in Figure 8a, there is a broad emission band centered at about 600 nm for all the EDTA-CdSe/CdS QDs systems added with various Cd^2+^ concentrations. Figure 8b shows the change trend of PL intensity of EDTA-CdSe/CdS system with the increase of Cd^2+^ concentration in the detected solution, and the fluorescence restoration increases continually in the concentration range of 0.1–90 µmol/L Cd^2+^ of ultrapure water.

Figure 8c,d show the linear fitting results of the data of *I*/*I*_0_ and Cd^2+^ concentration in two regions: 0.1–20 μmol/L and 20–90 μmol/L, corresponding to the linear Equations (5) and (6), respectively. The correlation coefficients (R^2^) are all above 0.99, suggesting a good linear relationship between *I*/*I*_0_ and Cd^2+^ concentration. According to the Equation (7) [33], where *δ* is the standard deviation of blank measurements (*n* = 11) and *S* is the slope of calibration graph. The detection limit (LOD) was calculated to be 6 nmol/L.
(5)$$I/I_{0}=1.05536+0.03529C_{{\mathrm{Cd}}^{2+}}\;0.1\text{–}20\;\mu \mathrm{mol}/L$$
(6)$$I/I_{0}=0.6664+0.05869C_{{\mathrm{Cd}}^{2+}}\;20\text{–}90\;\mu \mathrm{mol}/L$$
(7)LOD=3δ/S

The selectivity of EDTA-CdSe/CdS QDs system to Cd^2+^ in an aqueous solution was evaluated in comparison with 13 other metal ions (K^+^, Na^+^, Mg^2+^, Ba^2+^, Al^3+^, Mn^2+^, Fe^3+^, Ca^2+^, Hg^2+^, Pb^2+^, Cu^2+^, Ag^+^, Zn^2+^) under the optimal fluorescence restoration conditions. In order to reflect the selectivity of the QD system to various impurities adequately, the concentration of these interfering ions is set to 500 µmol/L except for Zn^2+^, and both of the Zn^2+^ and Cd^2+^ concentrations are set to be 50 µmol/L. Figure 9 shows the effects of these interfering ions on fluorescence restoration efficiency of the EDTA-CdSe/CdS fluorescence probe. Cu^2+^, Ag^+^, Hg^2+^, and Pb^2+^ lead to fluorescence quenching of the system completely. They could be adsorbed on QDs surface and quenched PL due to electron transfer from QDs to Ag^+^, Cu^2+^, Hg^2+^, and Pb^2+^. In addition, a chemical displacement of surface Cd^2+^ by Hg^2+^, Cu^2+^, and Ag^+^ occurred due to the extremely low solubility of CuSe, HgSe, and Ag_2_Se. Their formation would cause the PL quenching by facilitating non-radiative electron/hole (e^−^/h^+^) annihilation for the QDs system [34,35,36]. Ni^+^, Fe^3+^, and Mn^2+^ also have a certain fluorescence quenching effect on the system. K^+^, Na^+^, Al^3+^, Ba^2+^, and Ca^2+^ have little effect on fluorescence. However, Cd^2+^ shows a significant fluorescence restoration effect on the system, and *I*/*I*_0_ increases by 3.6 times with the addition of Cd^2+^. However, Zn^2+^ also shows an obvious fluorescence restoration effect, and *I*/*I*_0_ increases by 2.0 times with the addition of Zn^2+^, likely owing to similar chemical properties between Zn^2+^ and Cd^2+^. Zn-mercaptan forming on the surface of QDs can lead to fluorescence restoration for the system [30]. In conclusion, the CdSe/CdS QDs system has a high selectivity for the detection of Cd^2+^ in an aqueous solution, but it is improper to detect Cd^2+^ in a solution containing Cd^2+^ and Zn^2+^.

### 3.5. Detection of Cd^2+^ in Tap Samples

The detection experiments of Cd^2+^ were conducted in tap water to evaluate the practicability and reliability of the EDTA-CdSe/CdS QDs fluorescent probe. The tap water is a living water from Nanning city of China. The solutions with three different concentrations of Cd^2+^ (10, 20 and 30 µmol/L) were introduced into the tap water, respectively, and the experimental results are shown in Table 1. It can be seen that the measured values are very close to the actual concentration of Cd^2+^ in the aqueous solutions, with the fluorescence restoration efficiency above 96% and the relative standard deviation (RSD) below 4%. It suggests that the inherent ions such as Na^+^, Ca^2+^, Mg^2+^, and Mn^2+^ in the tap water cannot constitute a barrier in the detection of Cd^2+^ for the EDTA-CdSe/CdS QDs fluorescent probe. Note that the actual Cd^2+^ concentration in the tap water was considered to be equal to the dosage of Cd^2+^ because it is not detectable in pristine tap water.

## 4. Conclusions

A CdSe/CdS QD fluorescent probe with core-shell structure was successfully synthesized via solution reaction method. EDTA was proved to be an efficient fluorescence quenching agent to realize the OFF function in the detection process of Cd^2+^ based on an OFF–ON mode. Under proper conditions, the fluorescence quenching efficiency reached 99.5% within 5 min. The fluorescence of the EDTA-CdSe/CdS system was restored more effectively when Cd^2+^ was immigrated in the solution system, realizing the ON function in the detection process. The relationship between fluorescence restoration efficiency and Cd^2+^ concentration was well expressed by two linear equations, which can be used for the accurate calculation of Cd^2+^ concentration in the 0.1–100 μmol/L range. The QD fluorescent probe shows a good selectivity for Cd^2+^ in the aqueous solution containing twelve kinds of interfering ions such as Na^+^, Ca^2+^, Mg^2+^, Mn^2+^, and so on. However, the presence of Zn^2+^ would cause serious interference for the detection of Cd^2+^ in an aqueous solution. Finally, the experimental results of Cd^2+^ detection in tap water further proved the practicability and reliability of the EDTA-CdSe/CdS QDs fluorescent probe.

## Figures and Tables

**Figure 1 toxics-10-00367-f001:**
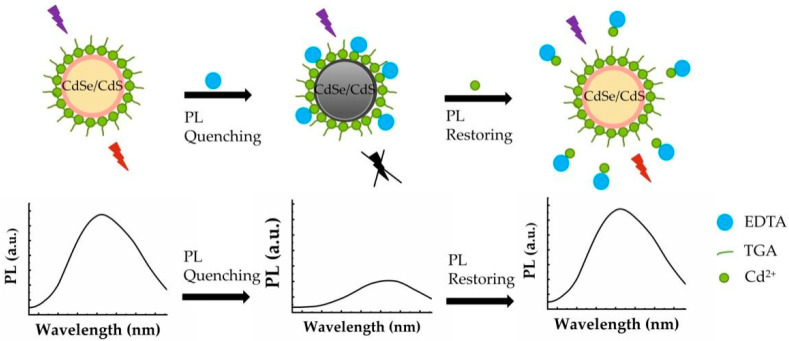
A schematic diagram of the detection of Cd^2+^ in an aqueous solution by CdSe/CdS QDs based on OFF–ON mode through absorption and desorption processes.

**Figure 2 toxics-10-00367-f002:**
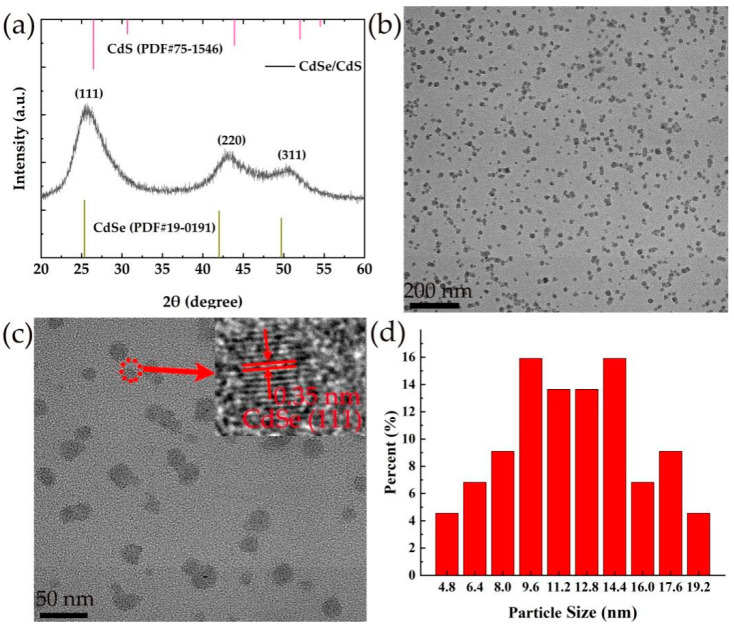
(**a**) XRD of CdSe/CdS QDs, (**b**,**c**) TEM images of CdSe/CdS QDs in different scale, and (**d**) size distribution of CdSe/CdS QDs.

**Figure 3 toxics-10-00367-f003:**
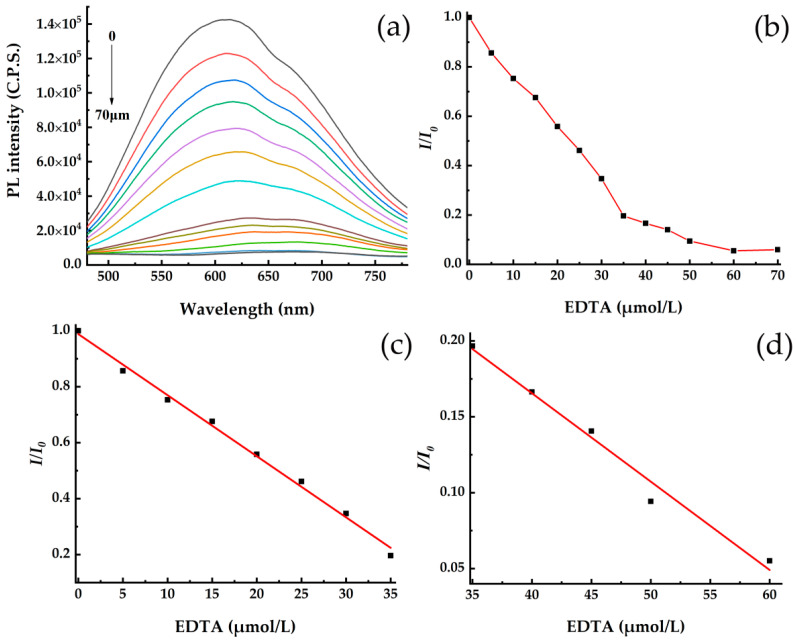
(**a**) Fluorescence spectra of the CdSe/CdS QDs system with various EDTA concentrations (0–60 µmol/L) in 300 µL volume, and (**b**) the relationship between fluorescence intensity ratio (*I*/*I*_0_) of CdSe/CdS QDs and EDTA concentration, (**c**) 0–35 µmol/L EDTA, and (**d**) 35–60 µmol/L EDTA.

**Figure 4 toxics-10-00367-f004:**
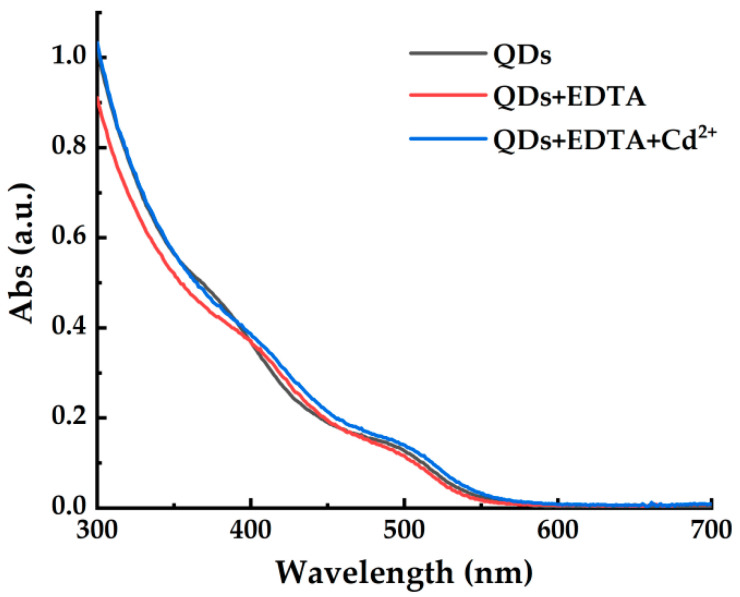
The absorption spectra of the systems of QDs, EDTA-QDs, EDTA-QDs+50 μmol/L Cd^2+^.

**Figure 5 toxics-10-00367-f005:**
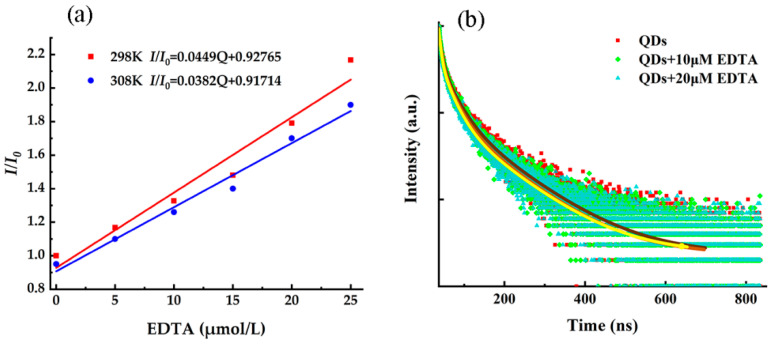
(**a**) Stern–Volmer plot for CdSe/CdS QDs system at 298 K and 308 K quenched by EDTA, and (**b**) fluorescence lifetime of the systems of QDs, QDs + 10 µmol/L EDTA, and QDs + 20 µmol/L EDTA.

**Figure 6 toxics-10-00367-f006:**
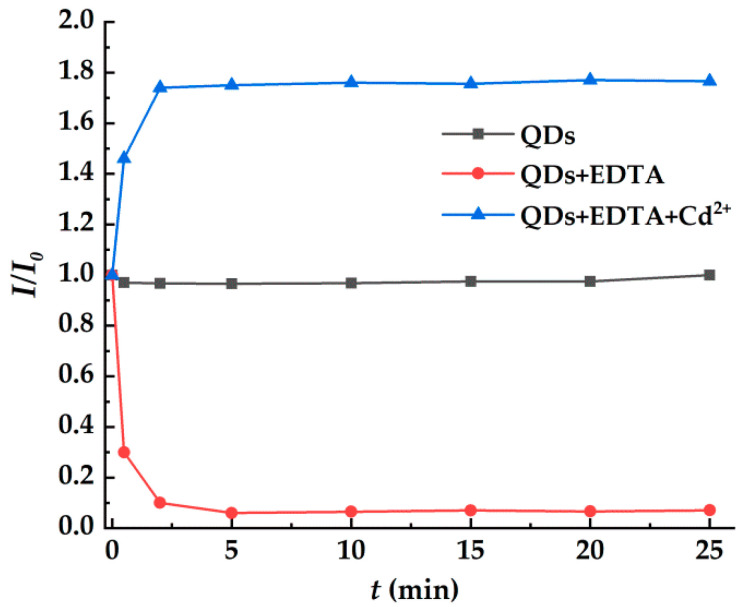
The change trend of *I*/*I*_0_ for the system of QDs, EDTA-QDs, EDTA-QDs + 20 μmol/L Cd^2+^ with incubation time (*t*).

**Figure 7 toxics-10-00367-f007:**
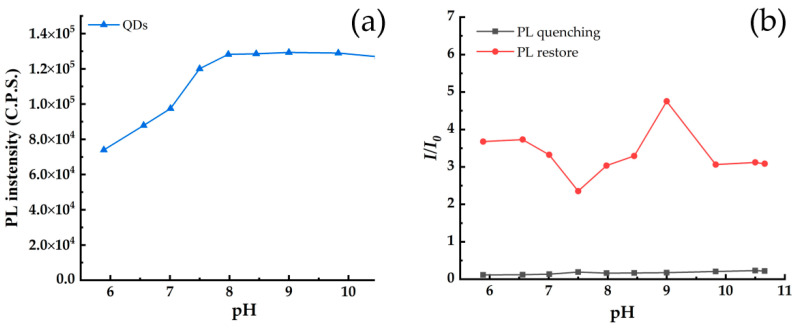
(**a**) The relationship between PL intensity of CdSe/CdS QDs and solution pH; (**b**) the effect of solution pH on the PL quenching and PL restoration.

**Figure 8 toxics-10-00367-f008:**
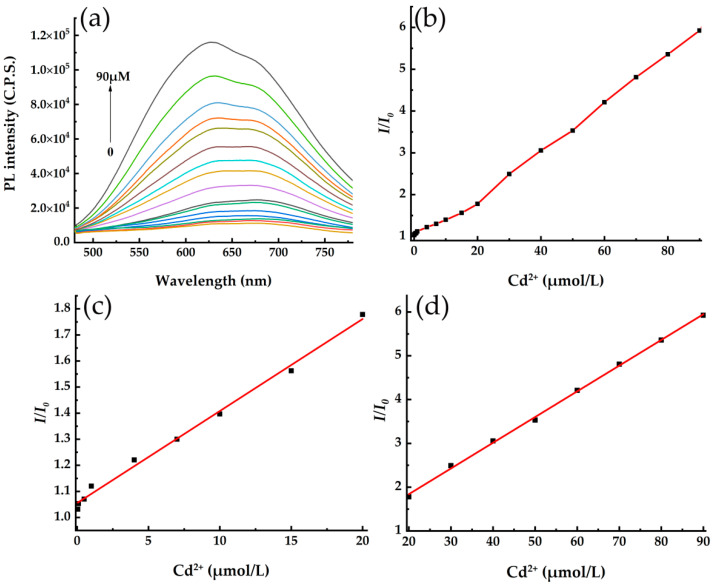
(**a**) The fluorescence spectra of EDTA-CdSe/CdS+Cd^2+^ QDs system with the various Cd^2+^ concentrations, and the relationship between fluorescence intensity ratio (*I/I*_0_) and Cd^2+^ concentration in the detected solution, (**b**) 0–90 µmol/L, (**c**) 0–20 µmol/L, and (**d**) 20–90 µmol/L.

**Figure 9 toxics-10-00367-f009:**
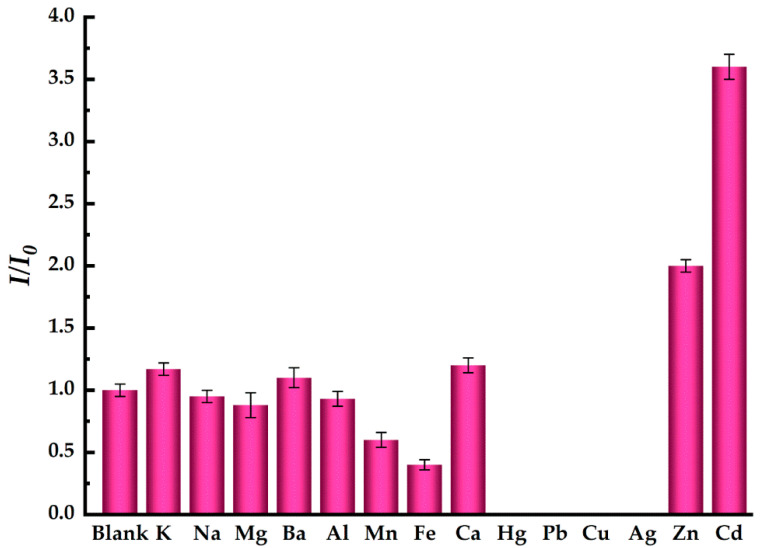
The effect of interfering ions on fluorescent restoration for the system of EDTA-CdSe/CdS QDs.

**Table 1 toxics-10-00367-t001:** A comparison between actual Cd^2+^ concentration and the measurasdfed values, and the fluorescence restoration efficiency and relative standard deviation for the detection of Cd^2+^ in the tap water.

Sample (No.)	Cd^2+^ Concentration (µmol/L)	ICP-MS Method	Proposed Method (µmol/L)	Recovery (%)	RSD (%, *n* = 3)
		(µmol/L)			
1	10	9.65	9.68	96.8	2.6
2	20	20.56	20.95	104.8	1.8
3	30	30.08	29.12	97.1	3.8
Mean value	/		/	99.6	2.8

## Data Availability

Data is contained within the article. The data presented in this study are available in this published article.

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
