# Peer review of "Detection of Cd2+ in Aqueous Solution by the Fluorescent Probe of CdSe/CdS QDs Based on OFF–ON Mode"

_toxics, 2022, doi:10.3390/toxics10070367_

Round 1
Reviewer 1 Report
The subject is interesting from my point of view, however the applied approach, methods and results suffers from several weaknesses and should be improved. My comments and advices are shown below. They should be taken into consideration. In my opinion the work may be publishable after major revision and incorporation of appropriate changes. 1. The observed fluorescence quenching should be analyzed with the use of Stern-Volmer equation. The fluorescence lifetime of CdSe/CdS QDs should be measured (if it is not known; check in literature) and then bimolecular quenching rate constant should be determined. The obtained value would be an evidence of appropriate mechanism of fluorescence quenching (dynamic or static). Alternatively, temperature experiments can be performed. 2. Authors present the fluorescence emission spectra of CdSe/CdS QDs in the presence of increasing amounts of EDTA and then the fluorescence emission spectra of EDTA-CdSe/CdS QDs in the presence of Cd2+ ions. In my opinion the UV-Vis spectra under the same conditions should be additionally recorded and presented. 3. How can Authors explain the observed shift of the band in emission spectrum? 4. How can Authors explain the phenomenon that the fluorescence quenching slows down at higher EDTA concentrations? And analogously, that the fluorescence enhancement progresses at concentrations of Cd2+ ions higher than 20 uM? 5. In figure 3d there is no point for the concentration of EDTA equal to 70 uM (as it is mentioned in the caption). 6. In my opinion the terms in lines 184-185 and 202-203 are contradictory. 7. In figure 3a and 4a “um” should be corrected to “uM” or “umol/L”. 8. How can Authors explain that in case of some ions (for example Cu2+, Ag+, Mn2+, Fe3+) a further decrease of fluorescence intensity of EDTA- CdSe/CdS QDs is observed (contrary to Zn2+ and Cd2+)? 9. According to a method validation: the results for quantitative determination of Cd2+ ion should be compared with at least one other commonly used technique. 10. In the case of selectivity experiments Authors used various metal ions at concentrations equal to 500uM at pH 8.0. Was the potential hydrolysis considered? 11. The fluorescence intensity values were not corrected for inner-filter effects. Probably in the case of Cd2+ ions these effects are negligible, but it should be emphasized.
Author Response
- The observed fluorescence quenching should be analyzed with the use of Stern-Volmer equation. The fluorescence lifetime of CdSe/CdS QDs should be measured (if it is not known; check in literature) and then bimolecular quenching rate constant should be determined. The obtained value would be an evidence of appropriate mechanism of fluorescence quenching (dynamic or static). Alternatively, temperature experiments can be performed.
R: About the quenching mechanism of the QDs by EDTA, we followed your advice and supplemented temperature experiments and it is expressed as follows in the revised manuscript.
"In order to further explore the mechanism of QDs quenching by EDTA, temperature experiments and measurement of the fluorescence lifetimes were conducted. The results are shown in Figure 5. The fluorescence quenching data were analyzed by Stern-Volmer equation [29] (3),
where Ksv is the quenching constant, and [Q] is the quenching agent concentration.
As shown in Figure 5(a), when the temperature is 298 K and 308 K, the relationship between I0/I and Q conforms to the Stern-Volmer equation. The value of slope decreases with the increase of temperature, indicating that static quenching occurs between EDTA and QDs. As shown in Figure 5(b), the fluorescence lifetimes of QDs, QDs+10 µmol/L EDTA and QDs+20 µmol/L EDTA were 27.3 ns, 24.4 ns and 23.7 ns, respectively. The change range of the lifetime is not obvious, consistent with the characteristics of static quenching. When the temperature is 298 K and 308 K, the quenching constants Ksv can be calculated to be 4.49×104 L/mol and 3.82×104 L/mol. According to the formula [31] (4),
where, τ0 is the lifetime of CdSe/CdS QD, the bimolecular quenching rate constant (Kq) is 1.64×1012 L·mol/s and 1.39×1012 L·mol/s, respectively, much higher than the maximum dynamic quenching rate 2.0×1010 L·mol/s. Therefore, it is concluded that the mechanism of QDs quenching by EDTA is static quenching. "
[29] Mohammad-Rezaei, R.; Razmi, H.; Abdolmohammad-Zadeh, H. d-penicillamine capped cadmium telluride quantum dots as a novel fluorometric sensor of copper (II). Luminescence 2013, 28, 503-509.
[31] Jiang, C.Q.; Gao, M.X.; He, J.X. Study of the interaction between terazosin and serum albumin: Synchronous fluorescence determination of terazosin. Analytica Chimica Acta 2002, 452, 185-189.
(Lines 200-219)
Figure 5. (a) Stern-Volmer plot for CdSe/CdS QDs quenching by EDTA at different temperatures,and (b) fluorescence lifetime of QDs,QDs+10 µmol/L EDTA和QDs+20 µmol/L EDTA
- Authors present the fluorescence emission spectra of CdSe/CdS QDs in the presence of increasing amounts of EDTA and then the fluorescence emission spectra of EDTA-CdSe/CdS QDs in the presence of Cd2+ions. In my opinion the UV-Vis spectra under the same conditions should be additionally recorded and presented.
R: About the the Uv-Vis spectrum, it is expressed as follows in the manuscript.
"As a common metal chelating agent, EDTA will be chemically absorbed on the surface of QDs to chelate with Cd2+ sites of CdSe/CdS QD when it was added to CdSe/CdS QD solution, and a large area of optical active sites on the surface of these fluorescent CdSe/CdS QDs were masked, resulting in the fluorescence quenching. It caused a blue shift in the absorption spectrum, as shown in Figure 4. After the introduction of Cd2+, the absorption peak was red shifted, which was due to the partial leakage of the photon into the shell matrix [30]. ".
[30] Xu, H.; Miao, R.; Fang, Z.; Zhong, X. Quantum dot-based “turn-on” fluorescent probe for detection of zinc and cadmium ions in aqueous media. Anal. Chim. Acta. 2011, 687, 82-88.
(Lines 190-196)
Figure 4. Absorption spectrum of the system of QDs、EDTA-QDs、EDTA-QDs+50 μmol/L Cd2+
- How can Authors explain the observed shift of the band in emission spectrum?
R: EDTA will be chemically absorbed on the surface of QDs to chelate with Cd2+ sites of CdSe/CdS QD when it was added to CdSe/CdS QD solution. That may cause QDs to cluster [29].
[29] Mohammad-Rezaei, R.; Razmi, H.; Abdolmohammad-Zadeh, H. d-penicillamine capped cadmium telluride quantum dots as a novel fluorometric sensor of copper (II). Luminescence 2013, 28, 503-509.
(Lines 166-167)
- How can Authors explain the phenomenon that the fluorescence quenching slows down at higher EDTA concentrations? And analogously, that the fluorescence enhancement progresses at concentrations of Cd2+ions higher than 20 μM?
R: As for the phenomenon of fluorescence quenching became slow at a higher EDTA concentration, we failed to explain this phenomenon before. Thanks for your question. Some Cd2+ sites on the surface of CdSe/CdS QD had been occupied with the increasing concentration of EDTA, and the further combination between EDTA and CdSe/CdS QD becomes difficult, so thus the fluorescence quenching slows down. When the concentration of Cd2+ ion is higher than 20 μM, the fluorescence enhancement becomes faster. This is because the higher the concentration of Cd2+ ions, the easier it is to combinate with EDTA. We have clarified it in the revised manuscript.
(Lines 173-176)
- In figure 3d there is no point for the concentration of EDTA equal to 70 µM (as it is mentioned in the caption).
R: Thanks for your discovery. It was our negligence. When the concentration of EDTA is 70 μM, it does not conform to the linear relationship. I have corrected it to the words "35-60 µmol/L" in the caption of Figure 3 (d) of manuscript.
- In my opinion the terms in lines 184-185 and 202-203 are contradictory.
R: Thanks for your finding. The words "The quenching effect is not obvious when the concentration of EDTA is lower while the detection of Cd2+ was not accurate when the concentration of EDTA was excessive" have been corrected to "The quenching effect is not obvious when the concentration of EDTA is too low while the detection of Cd2+ was not accurate when the concentration of EDTA is excessive. "
(Lines 220-223)
- In figure 3a and 4a “μm” should be corrected to “μM” or “umol/L”.
R: The unit "μm" has been corrected to "μM" in the revised manuscript.
- How can Authors explain that in case of some ions (for example Cu2+, Ag+, Mn2+, Fe3+) a further decrease of fluorescence intensity of EDTA-CdSe/CdS QDs is observed (contrary to Zn2+and Cd2+)?
R: Thank you for your question. We explain it as follows in the revised manuscript.
"They could be adsorbed on QDs surface and quench PL of QDs due to electron transfer from QDs to Ag+, Cu2+, Hg2+ and Pb2+. In addition, a chemical displacement of surface Cd2+ by Hg2+, Cu2+ and Ag+ occurred due to the extremely low solubility of CuSe, HgSe, Ag2Se. The formation of them would cause the luminescence quenching by facilitating non-radiative electron/hole (e−/h+) annihilation for the QDs system [34-36]."
[34] Xia, Y.S.; Cao, C.; Zhu, C.Q. Two distinct photoluminescence responses of CdTe quantum dots to Ag (I). Journal of Luminescence 2008, 128, 166-172.
[35] Xie, H.Y.; Liang, J.G.; Zhang, Z.L.; Liu, Y.; He, Z.K.; Pang, D.W. Luminescent CdSe-ZnS quantum dots as selective Cu2+ probe. Spectrochimica Acta Part A: Molecular and Biomolecular Spectroscopy 2004, 60, 2527-2530.
[36] Susha, A.S.; Javier, A.M.; Parak, W.J.; Rogach, A.L. Luminescent CdTe nanocrystals as ion probes and pH sensors in aqueous solutions. Colloids and Surfaces A: Physicochemical and Engineering Aspects 2006.
(Line 307-311)
- According to a method validation: the results for quantitative determination of Cd2+ion should be compared with at least one other commonly used technique.
R: Thanks for your suggestions. We have compared the results of the proposed method with the ICP-MS method, and see Table 1 of the revised manuscript, please.
- In the case of selectivity experiments Authors used various metal ions at concentrations equal to 500uM at pH 8.0. Was the potential hydrolysis considered?
R: We are very sorry that we did not consider the problem of metal ion hydrolysis because we did not find the solution turbid after the addition of the interfering ion solution during the experimental operations.
- The fluorescence intensity values were not corrected for inner-filter effects. Probably in the case of Cd2+ions these effects are negligible, but it should be emphasized.
R: Thanks for your suggestions. We have emphasized in the lines 137-138.

Reviewer 2 Report
In this manuscript, the authors reported on the synthesis of CdSe/CdS quantum dots and their possible use as sensing material for the presence of Cd(II) ions in pure and tap water. The paper is well organized, reasonably well written and the results could be of interest (mainly) to the scientific community working on chemical sensing. However, some additional measurements and some revisions/corrections have to be considered before publication:
1) In lines 239-240 the authors wrote “…EDTA on the surface of CdSe/CdS QDs produced a shielding function for the optical absorption and emission…”. In order to assess any variation of the optical absorbance, the UV-vis spectra of the QDs must be reported for: a) as prepared QDs, b) QDs after interaction with EDTA and c) after subsequent interaction with Cd(II).
2) The strong quenching effect produced by Hg, Pb, Cu and Ag will certainly hamper detection of Cd in water samples where one or more of these metals are present. The authors should comment on this, try to explain the working mechanism of these interfering metals and report it in the sentences from 276 to 278.
3) The origin (City/area, etc…) of the tap water used in the present study should be mentioned as well as the producer and characteristics of ultrapure water
4) Some typos:
line 24 “deviation of was within” put the missing words
lines 43-44 “insufficient in selectivity and insensitivity” is not really appropriate for some of the mentioned methods, the terms should be clarified.
line 98 “has ever been” -> “has already been”
line 146 “These obvious widen peaks reflects…” -> “These obvious wide peaks reflect…”
line 169 “becomes slowly” -> “becomes slow”
Author Response
- In lines 239-240 the authors wrote “…EDTA on the surface of CdSe/CdS QDs produced a shielding function for the optical absorption and emission…”. In order to assess any variation of the optical absorbance, the UV-vis spectra of the QDs must be reported for: a) as prepared QDs, b) QDs after interaction with EDTA and c) after subsequent interaction with Cd(II).
R: About the the Uv-Vis spectrum, it is expressed as follows in the revised manuscript.
"As a common metal chelating agent, EDTA will be chemically absorbed on the surface of QDs to chelate with Cd2+ sites of CdSe/CdS QD when it was added to CdSe/CdS QD solution, and a large area of optical active sites on the surface of these fluorescent CdSe/CdS QDs were masked, resulting in the fluorescence quenching. It caused a blue shift in the absorption spectrum, as shown in Figure 4. After the introduction of Cd2+, the absorption peak was red shifted, which was due to the partial leakage of the photon into the shell matrix [30]."
[30] Xu, H.; Miao, R.; Fang, Z.; Zhong, X. Quantum dot-based “turn-on” fluorescent probe for detection of zinc and cadmium ions in aqueous media. Anal. Chim. Acta. 2011, 687, 82-88.
(Lines 190-196)
Figure 4. Absorption spectrum of the system of QDs、EDTA-QDs、EDTA-QDs+50 μmol/L Cd2+
- The strong quenching effect produced by Hg, Pb, Cu and Ag will certainly hamper detection of Cd in water samples where one or more of these metals are present. The authors should comment on this, try to explain the working mechanism of these interfering metals and report it in the sentences from 276 to 278.
R: Thank you for your question. We explain it as follows in the revised manuscript.
"They could be adsorbed on QDs surface and quench PL of QDs due to electron transfer from QDs to Ag+, Cu2+, Hg2+ and Pb2+. In addition, a chemical displacement of surface Cd2+ by Hg2+, Cu2+ and Ag+ occurred due to the extremely low solubility of CuSe, HgSe, Ag2Se. The formation of them would cause the luminescence quenching by facilitating non-radiative electron/hole (e−/h+) annihilation for the QDs system [34-36]. "
[34] Xia, Y.S.; Cao, C.; Zhu, C.Q. Two distinct photoluminescence responses of CdTe quantum dots to Ag (I). Journal of Luminescence 2008, 128, 166-172.
[35] Xie, H.Y.; Liang, J.G.; Zhang, Z.L.; Liu, Y.; He, Z.K.; Pang, D.W. Luminescent CdSe-ZnS quantum dots as selective Cu2+ probe. Spectrochimica Acta Part A: Molecular and Biomolecular Spectroscopy 2004, 60, 2527-2530.
[36] Susha, A.S.; Javier, A.M.; Parak, W.J.; Rogach, A.L. Luminescent CdTe nanocrystals as ion probes and pH sensors in aqueous solutions. Colloids and Surfaces A: Physicochemical and Engineering Aspects 2006.
(Line 307-311)
- The origin (City/area, etc…) of the tap water used in the present study should be mentioned as well as the producer and characteristics of ultrapure water
R: Thanks for your suggestion. The origin of tap water is Nanning, China,and the resistivity of ultra-pure water reaches 18 MΩ·cm (25℃), as described in the manuscript. (Lines 325-326, Line 96-97)
- Some typos:
line 24 “deviation of was within” put the missing words
lines 43-44 “insufficient in selectivity and insensitivity” is not really appropriate for some of the mentioned methods, the terms should be clarified.
line 98 “has ever been” -> “has already been”
line 146 “These obvious widen peaks reflects…” -> “These obvious wide peaks reflect…”
line 169 “becomes slowly” -> “becomes slow”
R: Thanks for your suggestion, these Typos has been revised. (Line 25, Lines 43-46, Line 99, Line148, Line 173)
In addition, in order to introduce heavy metal pollution more naturally, we added a sentence in the Introduction part in the revised manuscript, which was highlighted in red. "Water pollution includes heavy metal pollution [1-3] and radionuclide pollution [4,5]. "
[1] Zhang, X.; Zhang, M.; Liu, H.; Gu, J.; Liu, Y. Environmental sustainability: a pressing challenge to biological sewage treatment processes. Current Opinion in Environmental Science & Health 2019, 12, 1-5.
[2] Yan, F.; Niu, Z.G. Evaluation model of major heavy metals pollution factors in coastal waters and sediments. Desalination and Water Treatment 2019, 149, 335-340.
[3] Huang, X.S.; Zhang, R.J.; Cui, M.J.; Lai, H.J. Experimental Investigation on Bioremediation of Heavy Metal Contaminated Solution by Sporosarcina pasteurii under Some Complex Conditions. Water 2022, 14, 595.
[4] Zhang, H.; Li, C.; Chen, X.; Fu, H.; Chen, Y.; Ning, S.; Fujita, T.; Wei, Y.; Wang, X. Layered ammonium vanadate nanobelt as efficient adsorbents for removal of Sr2+ and Cs+ from contaminated water. J Colloid Interface Sci. 2022, 615, 110-123.
[5] Luo, X.; Zhang, G.H.; Wang, X.; Gu, P. Research on a pellet co-precipitation micro-filtration process for the treatment of liquid waste containing strontium. Journal of Radioanalytical and Nuclear Chemistry 2013, 298, 931-939.
Thank you again.
Your sincerely!
Feng Gao, Dengpeng Wang

Round 2
Reviewer 1 Report
The manuscript has been significantly improved. In my opinion it merits to be published in Toxics journal. In the final version Authors should change the unit of bimolecular quenching rate constant. It should be "L mol-1 s-1" instead of "L mol s-1".